# Antidepressants Drug Use during COVID-19 Waves in the Tuscan General Population: An Interrupted Time-Series Analysis

**DOI:** 10.3390/jpm12020178

**Published:** 2022-01-28

**Authors:** Ippazio Cosimo Antonazzo, Carla Fornari, Sandy Maumus-Robert, Eleonora Cei, Olga Paoletti, Pietro Ferrara, Sara Conti, Paolo Angelo Cortesi, Lorenzo Giovanni Mantovani, Rosa Gini, Giampiero Mazzaglia

**Affiliations:** 1Research Centre on Public Health (CESP), University of Milan-Bicocca, 20900 Monza, Italy; ippazio.antonazzo@unimib.it (I.C.A.); e.cei@campus.unimib.it (E.C.); p.ferrara5@campus.unimib.it (P.F.); sara.conti@unimib.it (S.C.); paolo.cortesi@unimib.it (P.A.C.); lorenzo.mantovani@unimib.it (L.G.M.); giampiero.mazzaglia@unimib.it (G.M.); 2Team Pharmacoepidemiology, Bordeaux Population Health Research Center, Inserm U1219, University of Bordeaux, F-33000 Bordeaux, France; sandy.robert@u-bordeaux.fr; 3Regional Agency for Healthcare Services of Tuscany, Epidemiology, 50141 Florence, Italy; olga.paoletti@ars.toscana.it (O.P.); rosa.gini@ars.toscana.it (R.G.); 4Value-Based Healthcare Unit, IRCCS Multi Medica, 20099 Sesto San Giovanni, Italy

**Keywords:** COVID-19, antidepressants, pharmacoepidemiology, healthcare administrative database, lockdown

## Abstract

In Italy, during the COVID-19 waves two lockdowns were implemented to prevent virus diffusion in the general population. Data on antidepressant (AD) use in these periods are still scarce. This study aimed at exploring the impact of COVID-19 lockdowns on prevalence and incidence of antidepressant drug use in the general population. A population-based study using the healthcare administrative database of Tuscany was performed. We selected a dynamic cohort of subjects with at least one ADs dispensing from 1 January 2018 to 27 December 2020. The weekly prevalence and incidence of drug use were estimated across different segments: pre-lockdown (1 January 2018–8 March 2020), first lockdown (9 March 2020–15 June 2020), post-first lockdown (16 June 2020–15 November 2020) and second lockdown (16 November 2020–27 December 2020). An interrupted time-series analysis was used to assess the effect of lockdowns on the observed outcomes. Compared to the pre-lockdown we observed an abrupt reduction of ADs incidence (Incidence-Ratio: 0.82; 95% Confidence-Intervals: 0.74–0.91) and a slight weekly decrease of prevalence (Prevalence-Ratio: 0.997; 0.996–0.999). During the post-first lockdown AD use increased, with higher incidence- and similar prevalence values compared with those expected in the absence of the outbreak. This pandemic has impacted AD drug use in the general population with potential rebound effects during the period between waves. This calls for future studies aimed at exploring the mid–long term effects of this phenomenon.

## 1. Introduction

The COVID-19 pandemic caused by the SARS-CoV-2 virus has resulted in more than 228 million verified cases of virus infections and over 4.6 million COVID-19 related deaths globally [1]. Italy was one of the first countries in Europe to be hit by the new virus [2,3,4]. The emergency situation, characterized by an exponential growth of cases with high fatality rates, forced the Italian government to impose the first national lockdown from 9 March 2020 to 15 June 2020 to stop the virus spread in the general population [5]. In the following summer months, restrictions have been eased and people were allowed to work and travel. This resulted in a second large COVID-19 wave with a higher number of cases and consequent implementation of new lockdown restrictions from November 2020 to the end of December 2020 [5]. The measures adopted during the two lockdowns encompassed social distancing, smart-working, quarantine, people movement restriction, closure of non-essential activities, and limitation of the physical interactions.

The impact of stressful events on mental health has been already documented in the literature [6,7]. In particular, during the first COVID-19 wave an increased number of psychiatric diagnoses such as post-traumatic stress disorder (PTSD), depression, and anxiety were reported [7,8,9,10,11,12,13,14]. Previous studies already investigated the impact of lockdown on antidepressant drug (AD) use. In particular, some studies reported an increased use of these medications during the first wave [15,16] that in some case persist also during the subsequent period of 2020 [17,18,19,20]. On the contrary, other studies found an initial reduction of AD use during the peak of the first wave followed by an increased use in the subsequent periods [21]. Finally, some authors reported a non-significant change in psychotropic medication dispensing, including ADs, during the outbreak [22]. To the best of our knowledge, evidence on the impact of lockdowns on AD use in Italy is still scant and deserves careful evaluation.

This study, therefore, aimed at assessing whether the lockdown measures implemented in Italy during COVID-19 waves influenced prevalence and incidence of AD use in the general population.

## 2. Materials and Methods

### 2.1. Data Source and Ethical Approval

The method used in this study have been already described in other publications aimed at assessing the impact of lockdown on other drug classes in the Tuscan general population [23,24]. In this observational retrospective study, the healthcare administrative database (HAD) of Tuscany was used, covering approximately 3.6 million inhabitants. The HAD contains data on healthcare services accesses in the regional area, reimbursed by the Italian Healthcare System (NHS) to all regional citizens. The HADs consulted for this study included: (i) the archive of demographic and administrative data of individuals living in the catchment area who receive national healthcare service assistance; and (ii) the pharmacy claim registry that provides information (i.e., dispensing date, number of packages, substance name, anatomical therapeutic chemical code [ATC] and defined daily dose [DDD]) on all community prescription reimbursed by the NHS [25]. Approval for the study was obtained from the “*Agenzia Regionale di Sanità della Toscana*” Internal Governance Board.

### 2.2. Study Cohorts

All individuals with at least one dispensing of ADs (ATC: N06A*) between 1 January 2018 and 27 December 2020 were selected. The date of the first AD dispensing in the period was considered as the index date in the study. Subjects registered in the HAD less than 1 year before the index date were excluded from the cohort. Each patient was followed from the index date until 27 December 2020 or death/emigration, whichever came first.

### 2.3. Exposure Definition

During follow-up, patients were considered exposed to ADs when they claimed an AD dispensing of the same drug class (selective serotonin reuptake inhibitors—SSRIs, other antidepressants and tricyclic antidepressants). They were considered exposed until the end of duration of their last consecutive AD dispensing (treatment episode). We considered a dispensing as consecutive if it was claimed within the duration of the prior dispensing plus a 30-day grace period, which was added to account for irregular refill patterns and minor non-compliance. The duration of each dispensing was calculated by dividing the total quantity of active substance dispensed by the relevant defined daily dose (DDD) [26,27]. The number of treatment episodes was used to estimate the number of patients with incidental or prevalent AD use in each week of the study period. Patients switching therapy or using concomitantly more than one AD in the week, were counted as just one user.

### 2.4. Outcomes and Definitions

The outcomes of interest were weekly prevalence and incidence of AD use during the observation period. In particular, the weekly prevalence of AD use was calculated by dividing the weekly number of patients under AD treatment by the number of inhabitants living in the region at 1 January of each corresponding calendar year as reference population [28]. Similarly, the weekly incidence of AD use was estimated by including only patients who started an incident treatment episode at the numerator and the same reference population. For each treatment episode, we classified the patient as incident if he/she did not have an AD dispensed in the year prior the starting date of the treatment episode. During the study period, patients might have had more than one incident treatment episode.

### 2.5. Statistical Analysis

First, we estimated the weekly prevalence and incidence of ADs use in each segment of the study period: pre-lockdown period from 1 January 2018 to 8 March 2020 (114 weeks), the first lockdown from 9 March 2020 to 15 June 2020 (14 weeks), the post-First lockdown from 16 June 2020 to 15 November 2020 (22 weeks) and the initial period of second lockdown from 16 November 2020 to 27 December 2020 (6 weeks). Second, the impact of lockdown restrictions on the study outcomes was assessed by performing an interrupted time series (ITS) analysis with a quasi-Poisson generalized additive model [29,30,31]. The model included the weekly count of the observed outcome as response variable (Y) and the reference population as offset variable to transform the count outcome in incidence or prevalence. The fitted model was of the following form:Log [E(Y_i_)] = β_0_ + f (week_i_) + β_1_I (holiday_i_) + β_2_I (First lockdown_i_) + β_3_ (First lockdown week_i_) + β_4_I (Post-First lockdown_i_) + β_5_ (Post-First lockdown week_i_) + β_6_I (Second lockdown_i_) + β_7_ (Second lockdown week_i_).

The model included a non-linear function of the week (f(week_i_), spline function) and a dummy holiday indicator (0 = no, 1 = yes) to account for time trend and seasonality. Model coefficients were: β_0_, which represents the baseline level of outcome during the pre-lockdown period; β_2_, β_4_ and β_6_ which estimate the level change during first, post and second lockdown, respectively and β_3_, β_5_ and β_7_ which estimate the trends/slopes of the time series during the aforementioned periods [31,32]. Negative β_i_ coefficients indicate reduction over the corresponding period. In the model, a level change means an abrupt effect of the intervention whereas a change in trend/slope represents a gradual change in the estimated outcome [31]. In this specific case, a level change indicated an immediate and sustained effect on drug use, whereas a trend/slope change imply that the use of the antidepressants changed gradually in the studied segment. Next, we also investigated a possible delayed effect of lockdown restrictions implementation or cancellation, using the delayed or “lagged” level and slope indicators for all segments.

Finally, we plotted graphs of the outcomes over time to illustrate trends. Statistical significance of the parameters and the goodness of fit of the model were used to choose the best model. Significance was defined as a *p*-value less than 0.05. The models were implemented separately for each outcome.

In this study both the data processing and data analysis were performed using the R studio software (version 4.0.2 Rstudio, PBC: Boston, MA, USA).

### 2.6. Patient and Public Involvement

The research was conceived and performed by public institutions. All authors are employed or collaborate with public institutions.

## 3. Results

### 3.1. Prevalence of Drug Use

As reported in Table 1 and Figure 1A, a variation of the ADs prevalence was observed during the COVID-19 pandemic. In particular, the ITS analysis suggested a slight increase of weekly prevalence of ADs use in the first week of the first lockdown (Prevalence Ratio: 1.01; 95% CI: 1.00–1.02) followed by a weekly reduction (0.997; 0.996–0.999) for the entire first lockdown segment. During the post-First lockdown segment we observed a level change indicating a rapid increase of AD prevalence (1.02; 1.01–1.03) in the first week and slope change (1.002; 1.001–1.003) indicating a weekly increase of the prevalence for the rest of the period (Table 1 and Figure 1A). During the initial phase of second lockdown, no significant variation in AD prevalence was observed (Table 1 and Figure 1A).

### 3.2. Incidence of Drug Use

In the study period, a variation in the trend of AD incidence was observed in the Tuscan general population across the different segments (Figure 1B). As shown in Table 2, individuals were mainly female and aged 40 years and older. Among the study ADs drug classes, the most used was the selective serotonin reuptake inhibitors (SSRIs), whereas the less used were tricyclic ADs (Figure A1). As reported in Figure 1B, a level change, indicating an abrupt reduction of the AD incidence was observed during the first week of the first lockdown segment (Incidence Ratio: 0.82; 95% CI: 0.74–0.91). At post-First lockdown a level change, which indicates a rapid increase of AD incidence (1.30; 1.16–1.45) in the first week, was shown. During the initial phase of second lockdown, no significant variation was observed (Figure 1B and Table 1).

## 4. Discussion

The findings of this observational descriptive study showed that an abrupt reduction of AD incidence and a gradual reduction of AD prevalence occurred in the general population during the first lockdown. This initial reduction was followed by new increase of both prevalence and incidence during the subsequent periods (post-first lockdown and second lockdown). In particular, the AD prevalence reverted to values similar to those registered in the pre-lockdown period following the secular trend. On the contrary, after the first-lockdown the AD incidence reached higher values compared with expected values in case of no lockdown measures implementation.

Our data on AD use during the first lockdown appeared to be consistent with those reported in previous studies from Italy and other European countries that analysed mental health services access during the first lockdown [33,34,35,36]. The first COVID-19 outbreak has disrupted these services in most countries, especially during the first weeks of lockdown implementation. This might have caused an abrupt reduction of new diagnosis of psychiatric diseases with consequent reduction of new treatment cycle initiation. In parallel, the slight increase of the AD prevalence observed around the first week of the first lockdown and its subsequent reduction during the following weeks might result by combination of two phenomena. In fact, it is possible to speculate that during the early phase of virus spread (end of February) stress, worries and fear may have been the precursor of clinical exacerbation of existing anxiety/depression symptoms with consequent re-initiation of AD treatment in patients who had previously used ADs. On the contrary, during the subsequent weeks of lockdown patients might have been more reluctant to seek treatment, following the pattern of behaviour already described in other medical fields [34]. It is also possible that during the COVID-19 outbreak physicians were less prone to prescribe ADs to these patients by preferring other alternatives such as benzodiazepines. This hypothesis is supported by findings from Balestri and colleagues that documented an increased use of benzodiazepines in the psychiatric setting during the first lockdown [37].

During the post-first lockdown period, an increased use of ADs was registered. In particular, our results showed that the AD prevalence returned to pre-lockdown levels, whereas the number of new AD treatments were higher during this period compared with expected values in case of no lockdown measures implementation. In this regard, it should be noted that during the last phase of the first-lockdown and subsequent periods, several digital solutions were implemented to guarantee medical assistance to psychiatry patients [38]. This might have represented a crucial event for medical/therapeutic assistance to patients who experienced anxiety/depression symptoms onset during the last part of the first lockdown and those who had symptoms exacerbation with consequent mental health problems [39,40]. Additionally, data on incidence of drug use might also suggest a potential rebound/mid-long term effect of lockdown. In fact, it is possible to speculate that the emergency situation might have contribute to development of mental problems in individuals without pre-existing mental health problems who were resilient during the early phase of lockdown [37,41]. In this regard, stressors such as fear of contagion, long quarantine, isolation and movement restriction, lack of information, financial loss, inadequate supplies and stigma, and loss of relatives due to COVID-19 might have contributed to the deterioration in psychological outcomes with consequent onset of anxiety, depression and related symptoms [42,43,44].

The mid/long term consequences of the under-diagnosis and under-treatment of anxiety and depression during the first lockdown are still under-investigation. Previous studies suggested that an increase in psychiatric disorders and suicidal ideations and attempts might appear several months after stressful events even if a temporary reduction might be observed during the emergency situation [45,46,47,48,49]. This highlights the need of continuous monitoring of patients with psychiatry problems under exceptional circumstances like future COVID-19 waves. From clinical point of view, our findings suggest the necessity to encourage new tools such as telemedicine in order to guarantee medical assistance to psychiatric patients. This could be crucial to reduce the risk of adverse events due to non-optimal drug treatment. Additionally, telemedicine could increase also the effect of non-pharmacological management of these patients contributing to the aforementioned events improvements.

Future research should be performed to investigate whether new users of ADs during the pandemic outbreak had different demographic and clinical characteristics compared with those who started an AD treatment in the pre-lockdown period. In addition, new studies are warranted to investigate the potential consequence associated with the observed variations in AD use in the study population.

### Strengths and Limitations

An important strength of this study is that data related to an unselected population from one of the largest Italian regions, which accounts for over over 3.5 million of inhabitants. This study has some limitations. First, in Italy both ADs and anxiolytic drugs (i.e., benzodiazepines) are indicated for the treatment of the mental health symptoms after traumatic events. However, in Italy only ADs are fully reimbursed by the NHS and thus captured in the HAD, while the cost of anxiolytics is totally charged to patients. As a consequence, data about benzodiazepines use is generally not available in the Italian HAD. Moreover, the study did not include information about drug indications because in Italy this information is not available in the HAD. Additionally, for the second lockdown we included only six data points, which is slightly lower than the minimum number of points (eight) required to have robust estimates, therefore the results on the second lockdown should be considered with caution. Finally, as per study design we were not able to ensure a causal correlation between COVID-19 waves and studied outcomes, although the observed changes in the ADs use occurred around the studied events.

## 5. Conclusions

In conclusion, we found a change in AD use during pandemic lockdowns in the general population. There was a reduction of both prevalence and incidence of AD use during the first lockdown that reversed during the post-first lockdown reaching higher value compared with the expected results observed in the counterfactual scenario. The potential under-treatment occurred during the pandemic waves might be associated with increased risk of negative events (i.e., suicidal attempts) occurrence in the general population. These findings might be used to implement strategies to ensure access to adequate mental healthcare services for vulnerable population during extreme circumstances such as lockdown periods.

## Figures and Tables

**Figure 1 jpm-12-00178-f001:**
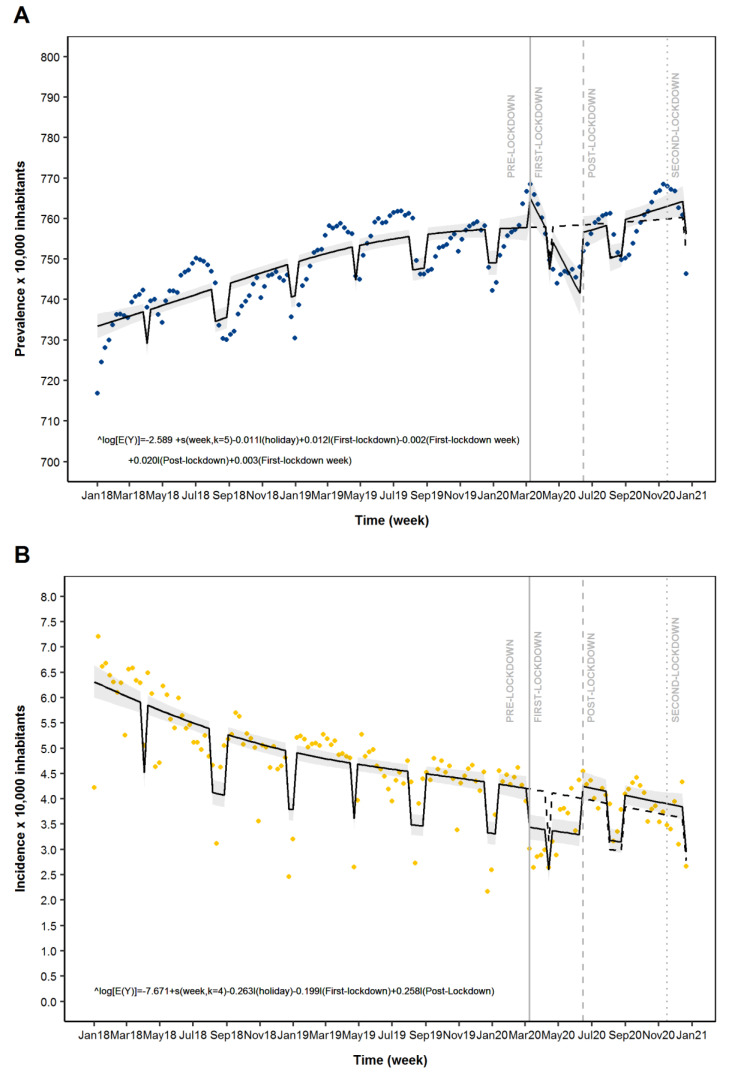
Time series analysis of antidepressant drugs (ADs) use across different periods: Pre-Lockdown, First-Lockdown, Post-First Lockdown and Second-Lockdown. Legend: Panel (**A**): Prevalence of AD use; Panel (**B**): Incidence of AD use. Blue dots: estimated prevalence of AD drug use; Orange dots: estimated incidence of AD drug use; solid line: predicted model based on estimated outcome; Grey zone: 95% CI of the predicted model; Dashed line: expected scenario in the absence of LMI.

**Table 1 jpm-12-00178-t001:** Time series analysis of antidepressants drug use across different time segments: Pre-Lockdown, First Lockdown, Post-First Lockdown and Second Lockdown.

Prevalence of AD Drug Use				
**Model Parameter ^1^**	**β**	**Prevalence Ratio**	**95% CI**	***p*-Value**
First Lockdown (1st week) ^3^	0.012	1.012	1.003–1.021	<0.05
First Lockdown (1st week) ^4^	−0.002	0.998	0.997–0.999	<0.001
Post-First Lockdown (1st week) ^3^	0.020	1.020	1.011–1.029	<0.001
Post-First Lockdown (1st week) ^4^	0.003	1.003	1.002–1.004	<0.001
**Incidence of AD Drug Use**				
**Model Parameter ^2^**	**β**	**Incidence Ratio**	**95% CI**	***p*-Value**
First Lockdown (1st week) ^3^	−0.199	0.819	0.742–0.905	<0.001
Post-First Lockdown (1th week) ^3^	0.258	1.295	1.160–1.445	<0.001

^1^—The GAM model was also corrected for holiday and a spline function of week (k = 5). ^2^—The GAM model was also corrected for month and a spline function of week (k = 4). ^3^—Level change; ^4^—Slope change.

**Table 2 jpm-12-00178-t002:** Demographic characteristics of incident AD users across different time segments: Pre-Lockdown, First Lockdown, Post-First Lockdown and Second Lockdown.

	Pre-Lockdown	First-Lockdown	Post-First Lockdown	Second-Lockdown
Number of weeks	114	14	22	6
Number of patients	175,563	14,684	27,695	6631
Sex, N (%)				
Female	112,024 (63.8)	8770 (59.7)	17,445 (63.0)	4109 (62.0)
Male	63,539 (36.2)	5914 (40.3)	10,250 (37.0)	2522 (38.0)
Age, N (%)				
18–29	10,315 (5.9)	898 (6.1)	1708 (6.2)	483 (7.3)
30–39	13,121 (7.5)	998 (6.8)	1825 (6.6)	490 (7.4)
40–49	23,245 (13.2)	1807 (12.3)	3285 (11.9)	840 (12.7)
50–59	28,598 (16.3)	2374 (16.2)	4184 (15.1)	1072 (16.2)
60–69	25,413 (14.5)	2169 (14.8)	4039 (14.6)	988 (14.9)
70–79	33,020 (18.8)	2689 (18.3)	5361 (19.4)	1168 (17.6)
80+	41,851 (23.8)	3749 (25.5)	7293 (26.3)	1590 (24)

## Data Availability

Not applicable.

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
