# Peer review of "Antidepressants Drug Use during COVID-19 Waves in the Tuscan General Population: An Interrupted Time-Series Analysis"

_jpm, 2022, doi:10.3390/jpm12020178_

Round 1

Reviewer 1 Report

Summary statement:

This study uses population-wide dispensing data to assess antidepressant use during COVID waves in Italy. While it includes data for the second lockdown, these data are likely insufficient to draw any robust conclusion. This is possibly why the discussion focuses on interpreting the first wave/lockdown. One suggestion is to restrict the analysis to the impacts during the first and post-first lockdown or update the analysis and the discussion incorporating further data points. Other clarifications and areas of concern are detailed below.

Major

L50-51: The introduction fails to recognise numerous previous studies assessing the impact of the COVID pandemic on antidepressant use and how this study adds to the previous literature. Some of these studies are:

  • Armitage R. (2021) Antidepressants, primary care, and adult mental health services in England during COVID-19. Lancet Psychiatry 8: e3.
  • Carr MJ, Steeg S, Webb RT, et al. (2021) Effects of the COVID-19 pandemic on primary care-recorded mental illness and self-harm episodes in the UK: a population-based cohort study. Lancet Public Health 6: e124-e135.
  • de Oliveira Costa, J., Gilles, M. B., Schaffer, A. L., Peiris, D., Zoega, H., & Pearson, S. A. (2021). Changes in antidepressant use in Australia: A nationwide analysis prior to and during the COVID-19 pandemic (2015-2021). medRxiv.
  • Jones CM, Guy GP, Jr. and Board A. (2021) Comparing actual and forecasted numbers of unique patients dispensed select medications for opioid use disorder, opioid overdose reversal, and mental health, during the COVID-19 pandemic, United States, January 2019 to May 2020. Drug Alcohol Depend 219: 108486
  • Milani, Sadaf Arefi, et al. "Trends in the Use of Benzodiazepines, Z-Hypnotics, and Serotonergic Drugs Among US Women and Men Before and During the COVID-19 Pandemic." JAMA network open10 (2021): e2131012-e2131012.
  • Stall NM, Zipursky JS, Rangrej J, et al. (2021) Assessment of Psychotropic Drug Prescribing Among Nursing Home Residents in Ontario, Canada, During the COVID-19 Pandemic. JAMA Intern Med.
  • Wolfschlag M, Grudet C, Håkansson A. Impact of the COVID-19 Pandemic on the General Mental Health in Sweden: No Observed Changes in the Dispensed Amount of Common Psychotropic Medications in the Region of Scania. Front Psychiatry. 2021 Dec 1;12:731297. doi: 10.3389/fpsyt.2021.731297. PMID: 34925084; PMCID: PMC8671297.
  • Uthayakumar S, Tadrous M, Vigod SN, Kitchen SA, Gomes T. The effects of COVID-19 on the dispensing rates of antidepressants and benzodiazepines in Canada. Depress Anxiety. 2021 Nov 29. doi: 10.1002/da.23228. Epub ahead of print. PMID: 34843627.

Please include in the text any restrictions on prescription durations imposed by regulatory bodies in Italy to prevent drug shortages (that could have affected your analysis). For example, people on chronic therapy coming back to refill prescriptions due to a shorter duration of treatment.

Methods

L93-94 Please clarify how individuals were classified as incident users (eg., without prior dispensing within 12 months)? Also, considering you follow up patients from 2018 to 2020, how did you account for repeated initiations (ie., a person with more than one episode meeting the definition of incident)?

L95-96 Please specify how you estimated the denominator for the incidence analysis (ie., did the authors remove from the denominator prevalent users?)

L102 – The initial period of the second lockdown in the methods differs from the one presented in the introduction

L113-122 Please clarify if observed data from the first waves of the COVID pandemic and post-first lockdown were used to estimate trends for subsequent periods. For instance, if you are estimating antidepressant initiation, "susceptible" people may have already initiated during "wave 1", so they couldn't initiate in "wave 2" or in the post-lockdown periods. Additionally, the further from "pre-pandemic" times, the less relevant the trends become to current practice.

Please comment on the ability to assess effects from the post-second wave, given you have few data points to get reasonable estimates (6 weeks). For instance, it would be helpful to include in table 1 the number of data points available to each estimation.

It is not clear why the authors report the results for the level change for the first week. Instead, they could have averaged the effects of each period assessed (that could balance the stockpiling effects possibly occurring in the first wave, for example) or included a pulse to represent the temporary increase in dispensing in March 2020 (See https://bmcmedresmethodol.biomedcentral.com/articles/10.1186/s12874-021-01235-8)

Maybe including an interpretation of what a Level change (i.e., immediate change that was sustained for the remainder of the period assessed) and a Slope change (i.e., immediate change occurring after the intervention) means would help in this regard.

L187-188 It is not clear if the rebound effect observed could be a consequence of stockpiling observed in the early days of the pandemic. This should be clearly articulated in the manuscript.

L271 How did the authors account for switches from one antidepressant class to another? Please include this information on the “Exposure definition” section

Author Response

Review_1

This study uses population-wide dispensing data to assess antidepressant use during COVID waves in Italy. While it includes data for the second lockdown, these data are likely insufficient to draw any robust conclusion. This is possibly why the discussion focuses on interpreting the first wave/lockdown. One suggestion is to restrict the analysis to the impacts during the first and post-first lockdown or update the analysis and the discussion incorporating further data points. Other clarifications and areas of concern are detailed below.

We thank the reviewer for the comments. We agree with the reviewer that we should have at least 8 points in the times series segments to draw any robust conclusion, but we didn’t exclude the second lockdown phase because we think the results may give a first indication. We revised the manuscript considering the reviewer’s issue. In the new version of the manuscript we carefully considered all raised points. 

Our point-by-point responses are provided below. 

Major

L50-51: The introduction fails to recognise numerous previous studies assessing the impact of the COVID pandemic on antidepressant use and how this study adds to the previous literature. Some of these studies are:

  • Armitage R. (2021) Antidepressants, primary care, and adult mental health services in England during COVID-19. Lancet Psychiatry 8: e3.
  • Carr MJ, Steeg S, Webb RT, et al. (2021) Effects of the COVID-19 pandemic on primary care-recorded mental illness and self-harm episodes in the UK: a population-based cohort study. Lancet Public Health 6: e124-e135.
  • de Oliveira Costa, J., Gilles, M. B., Schaffer, A. L., Peiris, D., Zoega, H., & Pearson, S. A. (2021). Changes in antidepressant use in Australia: A nationwide analysis prior to and during the COVID-19 pandemic (2015-2021). medRxiv.
  • Jones CM, Guy GP, Jr. and Board A. (2021) Comparing actual and forecasted numbers of unique patients dispensed select medications for opioid use disorder, opioid overdose reversal, and mental health, during the COVID-19 pandemic, United States, January 2019 to May 2020. Drug Alcohol Depend 219: 108486
  • Milani, Sadaf Arefi, et al. "Trends in the Use of Benzodiazepines, Z-Hypnotics, and Serotonergic Drugs Among US Women and Men Before and During the COVID-19 Pandemic." JAMA network open10 (2021): e2131012-e2131012.
  • Stall NM, Zipursky JS, Rangrej J, et al. (2021) Assessment of Psychotropic Drug Prescribing Among Nursing Home Residents in Ontario, Canada, During the COVID-19 Pandemic. JAMA Intern Med.
  • Wolfschlag M, Grudet C, Håkansson A. Impact of the COVID-19 Pandemic on the General Mental Health in Sweden: No Observed Changes in the Dispensed Amount of Common Psychotropic Medications in the Region of Scania. Front Psychiatry. 2021 Dec 1;12:731297. doi: 10.3389/fpsyt.2021.731297. PMID: 34925084; PMCID: PMC8671297.
  • Uthayakumar S, Tadrous M, Vigod SN, Kitchen SA, Gomes T. The effects of COVID-19 on the dispensing rates of antidepressants and benzodiazepines in Canada. Depress Anxiety. 2021 Nov 29. doi: 10.1002/da.23228. Epub ahead of print. PMID: 34843627.

 Please include in the text any restrictions on prescription durations imposed by regulatory bodies in Italy to prevent drug shortages (that could have affected your analysis). For example, people on chronic therapy coming back to refill prescriptions due to a shorter duration of treatment.

We thank the reviewer for the suggestion, which gave us the opportunity to update our introduction. In the new version of the manuscript, we updated the introduction section to include all suggested references.  We included the following sentence “Previous studies already investigated the impact of lockdown on antidepressants drug use. In particular, some studies reported an increased use of these medications during the first wave [15,16] that in some case persist also during the subsequent period of 2020 [17-20]. On the contrary, other studies found an initial reduction of ADs use during the peak of the first wave followed by an increased use in the subsequent periods [21]. Finally, some authors reported a non-significant change in psychotropic medication dispensing, including ADs, during the outbreak [22]. To the best of our knowledge, evidence on impact of lockdowns on ADs use in Italy is still scant and deserve carefully evaluation.”.

Then, we did not mention any drug restriction policy during the study periods because in Italy no restrictions were introduced.

Methods

L93-94 Please clarify how individuals were classified as incident users (eg., without prior dispensing within 12 months)? Also, considering you follow up patients from 2018 to 2020, how did you account for repeated initiations (ie., a person with more than one episode meeting the definition of incident)?

Thank you for your comment, it allowed us to better describe our methodological choices. As the reviewer observed, a patient might have more than one treatment episode in the study period. The classification in prevalent and incident cases was performed for each treatment episode, from the starting date of each episode. Therefore, patients might be considered incident users more than once in the study period. In the new version of the manuscript we better specify this aspect by including this sentence “For each treatment episode we classified the patient as incident if he/she did not present an ADs dispensing in the year prior the starting date of treatment episode. During the study period, patients might have had more than one incident treatment episode.”

L95-96 Please specify how you estimated the denominator for the incidence analysis (ie., did the authors remove from the denominator prevalent users?)

Thank you for your comment. In the analysis we used the entire general population as denominator for the incidence and prevalence estimates. We didn’t remove prevalent users from the denominator for incidence estimates, because we think the observed number of prevalent cases did not impact incidence estimates. Moreover, we decided to use an easy and immediate indicator for health care professionals.  In the new version of manuscript, we clarify this aspect as follows: “In particular, the weekly prevalence of ADs was calculated by dividing the weekly number of patients under ADs treatment by the number of inhabitants living in the region at 1st January of each corresponding calendar year as reference population [18]. Similarly, the weekly incidence of ADs use was estimated by including only patients who started an incident treatment episode at the numerator and the same reference population.”

L102 – The initial period of the second lockdown in the methods differs from the one presented in the introduction.

Thank you for the comment. We noted that the date for the second lockdown included in the introduction and methods were different, there was a typo. We updated the dates to be consistent in each part of the manuscript. Therefore, we updated the sentence in the introduction as follows: “This resulted in a second large COVID-19 wave with higher number of cases and consequent implementation of new lockdown restrictions since November 2020 to the end of December 2020.”.   

L113-122 Please clarify if observed data from the first waves of the COVID pandemic and post-first lockdown were used to estimate trends for subsequent periods. For instance, if you are estimating antidepressant initiation, "susceptible" people may have already initiated during "wave 1", so they couldn't initiate in "wave 2" or in the post-lockdown periods. Additionally, the further from "pre-pandemic" times, the less relevant the trends become to current practice.

In the ITS analysis we used the entire time series to estimate trend and slope changes for each segment, otherwise we couldn’t estimate changes in the series after adjusting for seasonality. “Susceptible” people initiating treatment during “wave 1”, contribute as prevalent users in post-lockdown periods or “wave 2” if they continued to be treated (no interruption of treatment of one year). Otherwise, if they stopped the treatment, they will contribute as no users, or newly incident users (if conditions are satisfied).

Please comment on the ability to assess effects from the post-second wave, given you have few data points to get reasonable estimates (6 weeks). For instance, it would be helpful to include in table 1 the number of data points available to each estimation.

In the new version of the  manuscript we added the number of weeks for each time segment in Table 2 and in the method section “First, we estimated the weekly prevalence and incidence of ADs use in each segment of the study period: pre-lockdown period from January 1st, 2018 to March 8th, 2020 (114 weeks), the first lockdown from March 9th, 2020 to June 15th, 2020 (14 weeks), the post-First lockdown from June 16th, 2020 to November 15th, 2020 (22 weeks) and the initial period of second lockdown from November 16th, 2020 to December 27th, 2020 (6 weeks).”. 

As regard the limited number of points included in the second wave, we agree with the reviewer. Therefore, we included this aspect in the limitations paragraph in order to specify that although it is possible to perform the analysis by using less than 8 points the results should be interpreted with caution. Therefore, in the limitation paragraph we included the following sentence “Additionally, for the second lockdown we included only 6 data points, which is slightly lower than the minimum number of points (8 point) to have robust estimates, therefore the results on the second lockdown should be considered with caution.”.

It is not clear why the authors report the results for the level change for the first week. Instead, they could have averaged the effects of each period assessed (that could balance the stockpiling effects possibly occurring in the first wave, for example) or included a pulse to represent the temporary increase in dispensing in March 2020 (See https://bmcmedresmethodol.biomedcentral.com/articles/10.1186/s12874-021-01235-8)

Thank you for his/her observation. We didn’t average the effects of each period, because we wanted to evaluate changes in time (trends), in this study, we were not interested in an averaged effect. Then we didn’t account for a pulse because we didn’t observe an immediate return to baseline levels, but a gradually decrease. Moreover, we didn’t apply ARIMA models because we were able to limit autocorrelation, adjusting for seasonality in the model. 

Maybe including an interpretation of what a Level change (i.e., immediate change that was sustained for the remainder of the period assessed) and a Slope change (i.e., immediate change occurring after the intervention) means would help in this regard.

Thank you for the comment. In the new version of the manuscript we included the following sentence to better specify what slope and level changes could mean in our study “In this specific case, a level change indicated an immediate and sustained effect on drug use, whereas a trend/slope change imply that the use of the antidepressants changed gradually in the studied segment.”

L187-188 It is not clear if the rebound effect observed could be a consequence of stockpiling observed in the early days of the pandemic. This should be clearly articulated in the manuscript.

Thank you for the comments which gave us the possibility to better explain our hypothesis. We observed a potential rebound effect in the incidence time series and not in the prevalence series, so we think it is difficult to relate the stockpiling observed in the early days of the pandemic for prevalence to a possible rebound effect in incidence during post-lockdown phase. Therefore, in the new version of the manuscript we rephrase the sentence as follows “Additionally, data on incidence of drug use might also suggest a potential rebound/mid-long term effect of lockdown. In fact, it is possible to speculate that the emergency situation might have contribute to development of mental problems in individuals without pre-existing mental health problems who were resilient during the early phase of lockdown [35, 39]. In this regard, stressors such as fear of contagion, long quarantine, isolation and movement restriction, lack of information, financial loss, inadequate supplies and stigma, and loss of relatives due to COVID-19 might have contributed to the deterioration in psychological outcomes with consequent onset of anxiety, depression and related symptoms [40-42].”.

L271 How did the authors account for switches from one antidepressant class to another? Please include this information on the “Exposure definition” section

Thank you for the comment. In the new version of manuscript, we included more details on exposure assessment. In particular, we included the following sentence “During follow-up, patients were considered exposed to ADs when they claimed an AD dispensing of the same drug class (Selective Serotonin Reuptake inhibitors-SSRIs, others antidepressants and tricyclic antidepressants). They were considered exposed until the end of duration of their last consecutive AD dispensing (treatment episode). We considered a dispensing as consecutive if it was claimed within the duration of the prior dispensing plus 30-days of grace period, which was added to account for irregular refill patterns and minor non-compliance. The duration of each dispensing was calculated by dividing the total quantity of active substance dispensed by the relevant Defined Daily Dose (DDD) [24,25]. The treatment episode was used to estimate the number of patients with incident or prevalent ADs use in each week of the study period. Patients switching therapy or using concomitantly more than one AD in the week, contributed as just one user.”.

Reviewer 2 Report

Very interesting and thought provoking study with a discussion generally covering all the possible conclusions. It seems counter-intuitive that AD needs in this big population would fall with the onset of the lockdown but it could be like a war, where people are energised to survive; but of course there is the issue of the medical services diverting attention to more important issues. There are a few mistakes I saw:

  1. line 43 /44 I assume they mean 2020 and not 2021
  2. line 110 equation first.
  3. What is ITS on line 140?

Author Response

Reviwe_2

We thank the reviewer for the comments to our manuscript. In the new version of the manuscript we addressed all raised concerns.

Very interesting and thought provoking study with a discussion generally covering all the possible conclusions. It seems counter-intuitive that AD needs in this big population would fall with the onset of the lockdown but it could be like a war, where people are energised to survive; but of course there is the issue of the medical services diverting attention to more important issues. There are a few mistakes I saw:

  1. line 43 /44 I assume they mean 2020 and not 2021

Amended as suggested.

  1. line 110 equation first.

Amended as suggested.

  1. What is ITS on line 140? 

Thank you for the comment. We included the meaning in the methods section, when we mentioned ITS for the first time. We can include the meaning also in the line140, but we are not sure that this is in line with journal guidelines.

Reviewer 3 Report

have a look at attached file

Author Response

Review_3

We thank the reviewer for the comments to our manuscript which gave us the possibility to improve it. In the new version of the manuscript we carefully considered all raised points. 

Our point-by-point responses are provided below. 

General

Authors explore in their manuscript ‘Antidepressants drug use during COVID-19 waves in the general population: an interrupted time-series analysis’ the use of antidepressant drugs during COVID-19 waves in the Tuscan general population. The use of ADs related to other diseases is understudied, however some minor work should be done before this manuscript can be published.

Title

Consider whether you would change the title <Antidepressants drug use during COVID-19 waves in the general population: an interrupted time-series analysis> into ‘Antidepressants drug use during COVID-19 waves in the Tuscan general population: an interrupted time-series analysis’

(the reader then does not mix this up with the Italian general population)

We agree with the reviewer, we modified the title as suggested.

Abstract

Background

Please add when both lockdowns (in Italy) were present.

Thank you for the comment. In the new version of abstract we have added the dates for each studied segment. Therefore, we have added the following sentence “The weekly prevalence and incidence of drug use were estimated across different segments: pre-lockdown (January 1st, 2018-March 8th, 2020), first lockdown (March 9th, 2020-June 15th, 2020), post-First lockdown (June 16th, 2020-November 15th, 2020) and second lockdown (November 16th, 2020-December 27th, 2020).”.

Methods

-

Results

-

Conclusion

-

Key words

Please add the word <lockdown>

Amended as suggested.

Introduction

Please change <Italy was one of the first country in Europe> into ‘Italy was one of the first countries in Europe’

Please change <characterized by exponential growth> into ‘characterized by an exponential growth’

Amended as suggested.

Methods

Sample

-

Measures

Please explain <All individuals with at least one dispensing of ADs (ATC: N06A*) between January 1st, 2018 and 27th December 2020 were selected>. Should I understand from this sentence that you selected the patients with one dispensing of ADs, and not that they were using ADs when you stopped collecting data (27th December 2020)? A GP could try just once AD use, and then stop prescription? Would such a person then be ‘in’?

We thank the reviewer for the comment, which gave us the opportunity to clarify the sentence. In the study, we included all individuals who hypothetically were under-treatment in the observed period. Therefore, it is possible that these patients were already under treatment or started a new treatment between 2018-2021. Additionally, it is possible that some patients had only 1 dispensing in the observed period, in this case the patients accounted for the period that we supposed to be under-treatment and then were considered as no under treatment. We hope our clarifications completely satisfy the reviewer’s concern. 

Statistical analyses

Please rewrite this section: First, we … . Then (Second), we … . Next, we … . Finally, we … . The readership easier grabs what you did and in which order the Results will be shown.

Amended as suggested.

Results

-

Discussion

Please change <in psychiatric setting> into ‘in a psychiatric setting

Amended as suggested.

Please add the heading:

Strengths and limitations

Mention strengths.

How do you know that people didn't use ADs (for them ‘for free’) when in fact indicated for benzodiazepines (not ‘for free’)?

We would like to thank the reviewer for the comment. In the new version of manuscript, we have separated the strengths and limitations paragraph from the rest of the discussion and have updated the section as follows

“Strengths and limitations

An important strength of this study is that data related to an unselected population from one of the largest Italian region which account for over over 3.5 million of inhabitants.  

This study has some limitations. First, in Italy both ADs and anxiolytic drugs (i.e., benzodiazepines) are indicated for the treatment of the mental health symptoms after traumatic events. However, in Italy only ADs are fully reimbursed by the NHS and thus captured in the HAD, while the cost of anxiolytics is totally charged to patients. As a consequence, data about benzodiazepines use is generally not available in the Italian HAD. Moreover, the study did not include information about drug indications because in Italy this information is not available in the HAD. Additionally, for the second lockdown we included only 6 data points, which is slightly lower than the minimum number of points (8 point) to have robust estimates, therefore the results on the second lockdown should be considered with caution. Finally, as per study design we were not able to ensure a causal correlation between COVID-19 waves and studied outcomes, although the observed changes in the ADs use occurred around the studied events.”.

As regard the question related to benzodiazepines use, unfortunately we did not have indications for drug use in the HAD, we have included also this point in the limits of the study “Moreover, the study did not include information about drug indications because in Italy this information is not available in the HAD.”.

Recommendations

Please add a paragraph on what your findings mean for practice, and what they mean for future research.

Conclusion

Please change <In conclusion, this study found change> into ‘In conclusion, we found a change’.

Amended as suggested.

Tables, Figures

-

References

-

Reviewer 4 Report

1. The study presents the results of original research.

2. Results reported have not been published elsewhere.

3. Experiments, statistics, and other analyses are performed to a high technical standard and are described in sufficient detail.

4. Conclusions are presented in an appropriate fashion and are supported by the data.

5. The article is presented in an intelligible fashion and is written in standard English.

6. The research meets all applicable standards for the ethics of experimentation and research integrity.

7. The article adheres to appropriate reporting guidelines and community standards for data availability.

Author Response

We are glad the reviewer judged our paper interesting and well written. We thank again the reviewer for his/her comments.

Reviewer 5 Report

The study could be interesting, but the results show are weak. Only a decrease in the prevalence of ad drug use, and the characteristics of these ADs. It is only descriptive and not very relevant for a journal with such a high impact factor.

I think this article should be submitted to a public health journal with a smaller international impact

Author Response

We greatly appreciate the time and efforts that the reviewer expended in reading our paper. We regret that the reviewer suggested to submit the study to another journal because we believe that it is in line with journal’s aims and scope – which include database exploitation for public health purposes, of utmost importance to tackle COVID-19 health emergency. Additionally, we believe that information from our study might be interesting for journal readers. We thank again the reviewer for the feedback and we hope that we the new verion of the manuscript he/she will change his/her opinion about it.

Round 2

Reviewer 1 Report

I was unable to revise the second version of this manuscript. It seems the authors have uploaded an old version since changes mentioned in the letter of reply were not performed in the manuscript text (eg, changes in the introduction).

Reviewer 5 Report

This version of the article is better in this form and seems to me acceptable for publication  

Round 3

Reviewer 1 Report

The authors have adequately addressed all the comments.